# Weakly Supervised Detection of Marine Animals in High Resolution Aerial Images

**Paul Berg** [1] , **Deise Santana Maia** [2] **, Minh-Tan Pham** [1,*] **and Sébastien Lefèvre** [1]

[1] Institut de Recherche en Informatique et Systèmes Aléatoires (IRISA), UMR 6074, Université Bretagne Sud, F-56000 Vannes, France; paul.berg@univ-ubs.fr (P.B.); sebastien.lefevre@univ-ubs.fr (S.L.)

[2] Centre de Recherche en Informatique, Signal et Automatique de Lille (CRIStAL), UMR 9189, Université de Lille, F-59000 Lille, France; deise.santanamaia@univ-lille.fr

[*] Correspondence: minh-tan.pham@univ-ubs.fr

**Abstract:** Human activities in the sea, such as intensive fishing and exploitation of offshore wind farms, may impact negatively on the marine mega fauna. As an attempt to control such impacts, surveying, and tracking of marine animals are often performed on the sites where those activities take place. Nowadays, thank to high resolution cameras and to the development of machine learning techniques, tracking of wild animals can be performed remotely and the analysis of the acquired images can be automatized using state-of-the-art object detection models. However, most state-of-the-art detection methods require lots of annotated data to provide satisfactory results. Since analyzing thousands of images acquired during a flight survey can be a cumbersome and time consuming task, we focus in this article on the weakly supervised detection of marine animals. We propose a modification of the patch distribution modeling method (PaDiM), which is currently one of the state-of-the-art approaches for anomaly detection and localization for visual industrial inspection. In order to show its effectiveness and suitability for marine animal detection, we conduct a comparative evaluation of the proposed method against the original version, as well as other state-of-the-art approaches on two high-resolution marine animal image datasets. On both tested datasets, the proposed method yielded better F1 and recall scores (75% recall/41% precision, and 57% recall/60% precision, respectively) when trained on images known to contain no object of interest. This shows a great potential of the proposed approach to speed up the marine animal discovery in new flight surveys. Additionally, such a method could be adopted for bounding box proposals to perform faster and cheaper annotation within a fully-supervised detection framework.

**Keywords:** marine animal monitoring; anomaly detection; deep learning; weakly supervised learning; convolutional neural networks

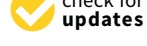

## 1. Introduction

With the ever-growing exploitation of marine natural resources, surveying human activities in the sea has become essential [1]. Activities, such as the installation of offshore wind farms and intensive fishing, should be closely monitored, as they can have a serious impact on the marine mega fauna. For instance, the noise produced during the different phases of an offshore wind farm development, including the site survey, the wind farm construction and the deployment of turbines, can potentially lead to various levels of physical injury, physiological, and behavioral changes in mammals, fish, and invertebrates [2–5]. In order to ensure that such human activities can take place without harming the marine ecosystem, different surveillance approaches have been adopted in the past years.

Nowadays, aerial surveys are among the standard non-invasive approaches for tracking the marine mega fauna [6–10]. Those surveys consist of flight sessions over the sea, during which environmental specialists are able to remotely observe the marine animals (e.g., seabirds, mammals, and fish) that emerge on the surface. In parallel, high resolution videos and photographs can be captured during the flight and, later, be used to validate

the observations made by the specialists. During a single flight session, thousands of aerial images, or a few hours of videos composed of thousands of frames, can be recorded. This makes the visual analysis of these data laborious and time consuming.

With the advance of deep learning techniques, such as convolutional neural networks (CNN), a natural direction towards optimizing marine mega fauna surveys is to automatize marine animal detection in aerial images using state-of-the-art methods for object detection [6–10]. Currently, the most efficient methods use variations of CNNs for feature extraction, and are trained in a supervised manner using lots of ground-truth bounding boxes. Hence, in order to use such methods, we still cannot skip the cumbersome task of analysing and annotating large amounts of data. On the other hand, using unsupervised and weakly supervised methods, we can benefit from all the available data without spending so much time on annotation. However, unsupervised models are still far behind supervised ones in terms of object detection performance. In this research, we aim to reduce the gap between the performances of supervised and unsupervised deep learning applied to object detection. This problem is tackled in the complex context of marine animal detection.

In this article, our main contributions are twofold: (1) a modification of the unsupervised anomaly detection method PaDiM (patch distribution modeling) [11], which we prove to be better adapted for marine animal detection than the original method; and (2) an evaluation of the proposed method and of other state-of-the-art approaches, namely PaDiM [11], OrthoAD [12], and AnoVAEGAN [13], on two high-resolution marine animal image datasets. Our codes are published and available online https://github.com/Pangoraw/MarineMammalsDetection (accessed on 4 January 2022).

## 2. Marine Animal Detection: Challenges and Current Solutions

The development of machine learning and, in particular, of deep learning methods in the past decade was boosted by an increasing computational power and by large amounts of available annotated datasets. Under favorable conditions, the accuracy of deep learning methods can even be similar to human's for some specific tasks, including pathology detection [14] and animal behavioral analysis [15]. For image classification on large datasets, such as ImageNet [16] (14,000,000+ annotated images), deep learning methods provide state-of-the-art results, reaching accuracy levels of over 90% [17]. Moreover, for object detection on large-scale image datasets, e.g., MS COCO [18] (300,000+ images with bounding box annotations belonging to nearly one hundred classes), deep learning also provides state-of-the-art results, though reaching human performance on such challenging scenarios is still an open problem.

In view of the success of deep learning in several computer vision tasks, extending the current state-of-the-art object detection methods for marine animal detection seems promising. On the one hand, a single session of an aerial survey over the sea surface can provide thousands of images with potentially several hundreds of animal instances, which, in theory, makes enough data to train a deep learning model. On the other hand, annotating this kind of data are a challenge for the following reasons:

1. Different animal species cannot be easily distinguished by untrained eyes, and, hence, annotations should be provided or at least validated by specialists. For instance, the dolphins of Figure 1a,b look very much alike, but they belong to different species: *Delphinus delphis* and *Stenella coeruleoalba*, respectively.

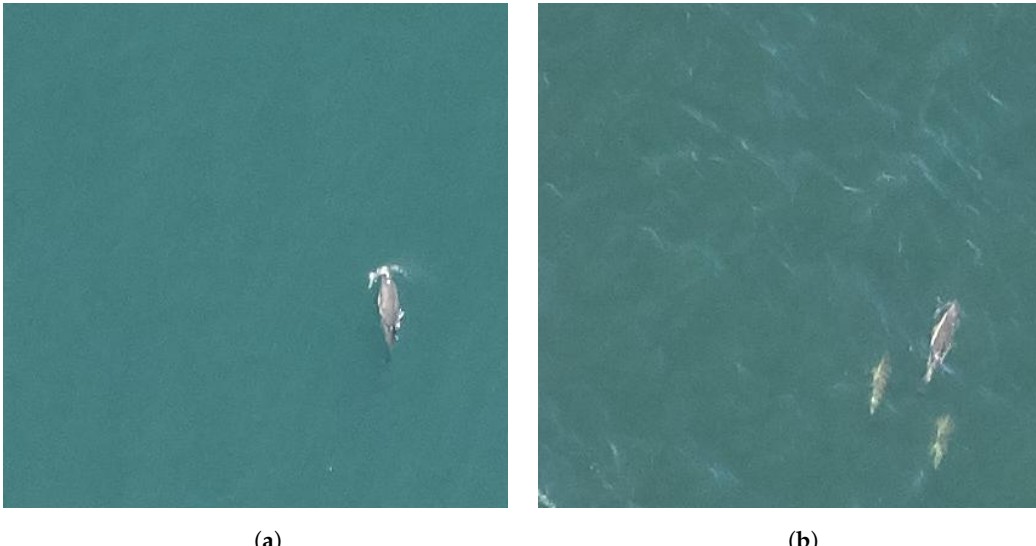

(**a**)　　　　　　　　　　　　　　　　　　　　　　(**b**)

**Figure 1.** Dolphins of the *Delphinus delphis* (**a**) and *Stenella coeruleoalba* (**b**) species.

2.　The appearance of marine animals changes as they swim deeper in the ocean, leading to ground-truth annotations with different confidence levels. For instance, in the images of Figure 2, the presence of dolphins of the *Delphinus delphis* species was confirmed by specialists, but lower confidence levels were assigned to those annotations due to their blurry appearance.

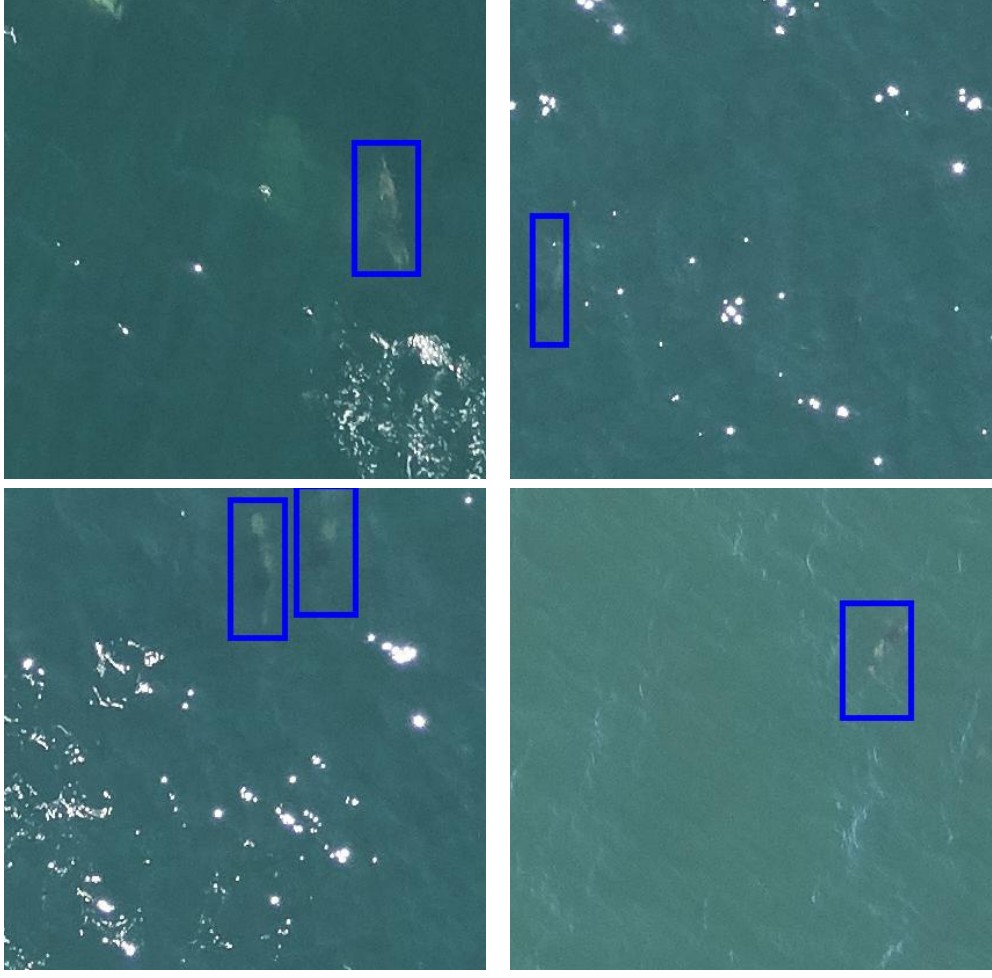

**Figure 2.** Ground-truth bounding boxes of dolphins with low confidence levels.

3.  Depending on the flight altitude, animal instances are so small that they can only be detected through their context. As an example, Figure 3 shows an image captured during a flight session of Ifremer (Institut Français de Recherche pour l'Exploitation de la Mer: https://wwz.ifremer.fr/ (accessed on 4 January 2022)). According to specialists, the bright dots inside the green bounding boxes probably correspond to marine animals, while that the ones inside the red box may be sun glitters. We can observe that this analysis is only possible by taking into consideration the proximity of each patch to the sun reflection.

4.  Although it is desirable to perform the flight sessions when the weather is favorable (no rain, not too much wind, and good visibility), it is not always possible due to other constraints, such as the availability of the pilot and other members of the crew. For that reason, waves crests and sun glitters, which may appear similar to animals (see Figure 4), are often visible in the images. Obtaining models which are robust to such kind of noise is one of the most difficult challenges in marine animal detection.

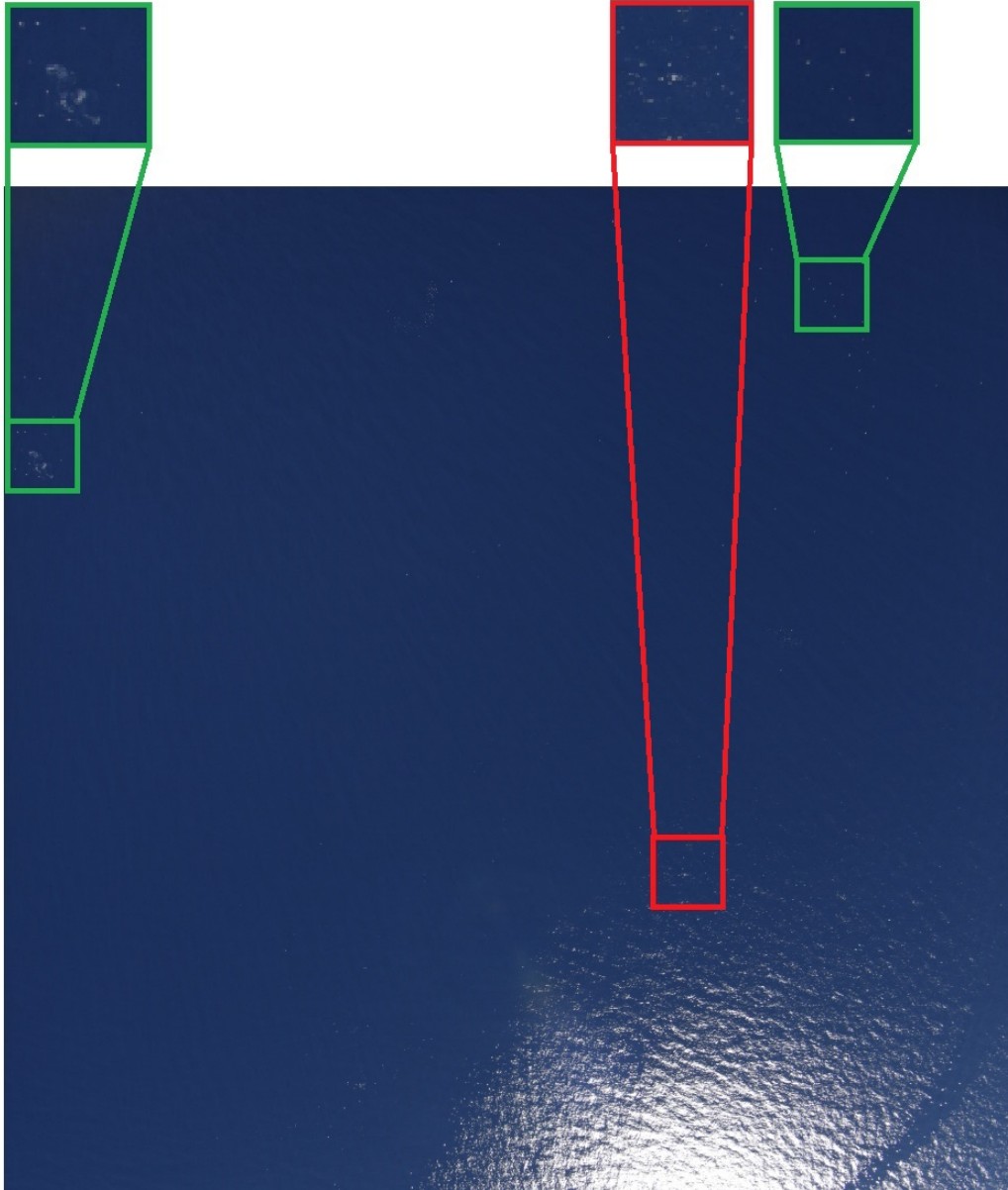

**Figure 3.** Image captured during an Ifremer flight session. The bright dots inside the green boxes may correspond to marine animals while that the ones inside the red box might be sun glitters.

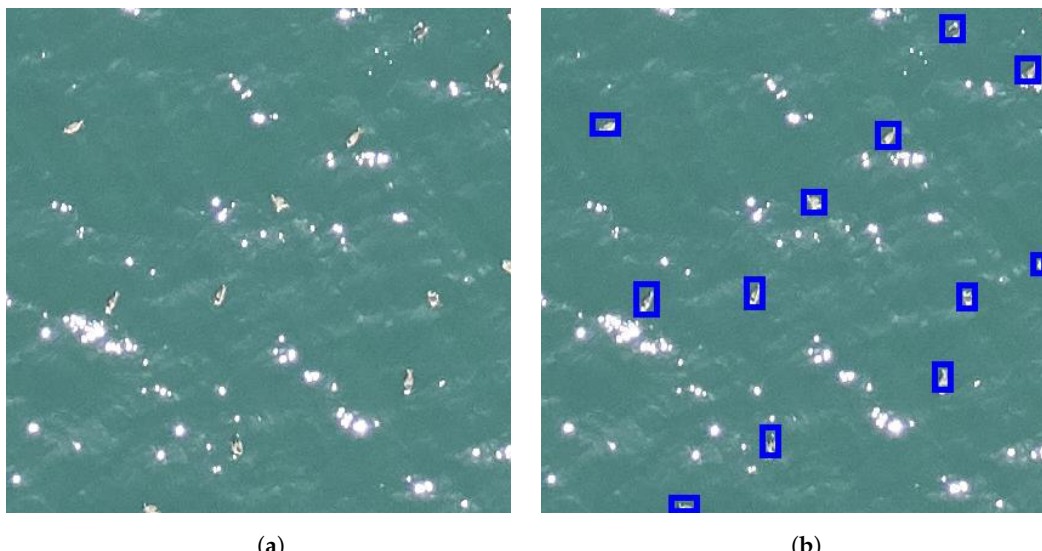

|  (**a**)  |  (**b**)  |

**Figure 4.** Original aerial image of marine birds (**a**) and its ground-truth bounding boxes (**b**).

Due to the complexity of detecting marine animals in those various scenarios, research studies in the literature often limit their scope to the detection of a single animal species [7,9] and/or to images with high density of animal instances [8,10]. For instance, in the early work of [7], the authors tackle the detection of dugongs in aerial images by combining an unsupervised region proposal method with a classification CNN. On their dataset, whose number of images was not provided, the best precision and recall scores were 27% and 80%, respectively. Similarly to [7], the authors of [9] targeted marine bird detection through a combination of an unsupervised region proposal with a classification CNN. Even though high accuracy scores (>95%) for their pre-trained CNN were reported, visual results presented in the paper show the difficulty of obtaining a model which is robust to sun glitters similar to the ones illustrated in Figure 4. In [6], the authors performed an end-to-end supervised detection of dolphins and stingrays in aerial images. Due to the high density and occlusion of animals in some areas, low average precision scores were obtained for both species: 30% and 35% for the detection of dolphins and of stingrays, respectively. In [8], both marine and terrestrial birds are targeted. As a novelty, the authors were able to boost the number of birds in their dataset by introducing samples of bird decoys. Using some of the state-of-the-art object detection models, including Faster R-CNN [19] or YOLOv4 [20], an average precision (AP) score of over 95% was reported on a set of positive samples, i.e., samples which contain at least one ground-truth bounding box.

In a more recent work on seabirds detection [10], efforts were made to reduce the manual workload required to obtain annotated training data. The authors trained a CNN to detect different species of seabirds, including terns and gulls, using only 200 training samples per class. To make up for the low number of training samples, prior-knowledge about the spatial distribution of birds was introduced during post-processing steps, which led to high precision and recall scores of approximately 90% for the most abundant class, but lower scores for the sparse classes. Though some of the methods reviewed above perform well on their dataset, they require some level of supervised labeling or some prior-knowledge about the distribution of the targeted animals. The literature on unsupervised and weakly-supervised methods for marine animal detection is still scarce, which motivated us to focus on weakly-supervised detection of different kinds of marine animals, such as turtles, birds, and dolphins, as described in the following sections.

### 3. Unsupervised and Weakly-Supervised Object and Anomaly Detection

The sparse distribution of marine animals makes it hard to gather sufficient data to train and test supervised models. Often, less than 5% of the images gathered during a flight survey will contain animals (see Section 5.1). The differences in appearance caused by the variations in animal depth shown in Figure 2 can also make it hard for supervised models to learn class-specific features. To better handle these constraints and to account for different weather conditions, we propose to train an object detector by applying anomaly localization techniques to sea images. By training on sea images without animals, our models require little to no-supervision compared to data-intensive supervised techniques. Classes with a very small number of training samples should offer comparable performance to that of other classes since the training data do not suffer from class imbalance. In our experiments, we focus on detection only. The classification of the detected animals will be left for future work.

Anomaly localization, as the name suggests, aims to localize the regions or area of pixels from an image that diverge from the "norm", where the norm is usually determined by image patterns (e.g., colors and textures) found in the training set. As a result, each pixel of an image is assigned an anomaly score. The goal is to detect all anomalous pixels that are different from the normal data present in the training set. A subset of this task is anomaly detection, where the goal is to classify whether an image contains an anomaly or not. We refer to an image without anomalies as a normal image and an image with anomalies as an anomalous image. Since marine animal detection requires predicting the precise location of an animal within an image, we focus on anomaly localization.

In the literature, most anomaly detection methods are proposed either in an industrial or in a medical context. The performance benchmarks are often made on the MVTec Anomaly Detection [21] (MVTec AD) dataset which contains a variety of textures and objects classes. The training set for each of these classes is composed of only normal images. A variety of methods already exists to localize anomalies in images, and some of them are reviewed below.

Reconstruction based methods train generative models to reconstruct the normal images from the training data by minimizing the reconstruction loss. The intuition is that anomalous samples will be poorly reconstructed and thus easy to detect by comparing the reconstruction with the original image. The most used models are autoencoders (AE) [22], variational autoencoders (VAE) [13,23] or adversarial autoencoders (AAE) [13,24]. Although easy to understand, generative models are sometimes able to reconstruct the anomalies even though they are not part of the training set, making the anomalies undetectable by standard dissimilarity measures computed from the original and reconstructed images. An anomaly can also lead to a failed reconstruction larger than the original anomaly making the precise anomaly localization impossible.

Deep embedding methods use the embedding vectors created by networks trained on other tasks to model the normal data. They can use a model pre-trained on another supervised dataset or on proxy tasks for a self-supervised training mode. To model the training data and detect embedding vectors that are anomalous, several methods have been proposed. Patch-SVDD [25] uses a proxy classifying task to encode the image and a Deep-SVDD [26] one-class classifier to classify the patch as either anomalous or normal. DifferNet [27] trains normalizing flows (NF) to maximize the likelihood of the training set and localizes anomalies by computing the gradient of the likelihood with regard to the input image. SPADE [28] compares the testing samples to the normal-only training set using a K-nearest neighbors retrieval on vectors created using a model pre-trained on supervised image classification. PaDiM [11] proposes to model each patch location using a Gaussian distribution and then use the Mahalanobis distance to compute the anomaly scores.

We experiment with both generative and embeddings based methods to reformulate the animal detection problem as an anomaly localization problem. Leveraging the fact that a majority of the recorded aerial imagery does not contain animals, we target marine animal detection models trained in a weakly-supervised setting.

## 4. Proposed Method for Weakly-Supervised Marine Animal Detection

Convolutional neural networks (CNN) pre-trained on supervised tasks have proven to be robust image feature extractors [28,29]. Their use in anomaly detection has already given interesting results in state-of-the-art benchmarks [11,12,30,31]. Since the benchmark datasets commonly used for anomaly detection from images are different from datasets available for marine mammals detection that can be made of thousands of images and involve a strong texture component, we propose to modify and adapt deep feature embedding methods to tackle the marine animals detection problem.

As first proposed in [28], to model the normal training set, the images are first encoded using a ResNet [32] model pre-trained on the ImageNet [16] dataset. To use different semantic levels, activations from the three intermediate layers are concatenated to create a feature map as used in [11,12,28]. Since this feature map is deep, the number of channels is often reduced using either random-dimensions selection [11] or a semi-orthogonal embedding matrix [12]. In practice, we found that using a semi-orthogonal embedding yields more consistent results because the random dimension selection requires to test multiple dimensions in order to find a good combination. The method [11] then models these normal feature maps using a Gaussian distribution for each patch location. During training, only a single forward pass is necessary to encode the training set and to compute the mean vectors and covariance matrices estimating the Gaussian distribution. Both can be computed online using the formulas in Equations (1) and (2):

$$\mu_{i,j} = \frac{1}{N} \sum_{k=1}^{N} x_{k,i,j} \tag{1}$$

$$\Sigma_{i,j} = \frac{1}{N-1} \left( \sum_{k=1}^{N} x_{k,i,j} x_{k,i,j}^{\top} - N \times (\mu_{i,j} \mu_{i,j}^{\top}) \right) + \epsilon I \tag{2}$$

where $x_{k,i,j}$ is the feature vector at location $i,j$ of the $k$th training sample and $N$ is the number of training samples. A regularization term $\epsilon I$, where $I$ is the identity matrix of corresponding size, is added to the covariance matrices for numerical stability for invariant patches as proposed in [11].

Once a Gaussian distribution has been estimated for each patch location, the anomaly score $s(x_{i,j})$ for each patch $x_{i,j}$ of a test image is computed using the Mahalanobis distance:

$$s(x_{i,j}) = \sqrt{(x_{i,j} - \mu_{i,j})^{\top} \Sigma_{i,j}^{-1} (x_{i,j} - \mu_{i,j})} \tag{3}$$

For the Gaussian distribution, the Mahalanobis distance is proportional to the square root of the negative log-likelihood. If the Gaussian distribution hypothesis is valid, detecting anomalous patches is similar to an out-of-distribution (OOD) samples detection process. With this method, the learnt distributions are depending on the patch location. This gives good performance on the MVTec AD dataset where the normal objects are always located at the same location in the image. This means that the model is not invariant to image transformations such as rotations and translations. However, such geometric transformations are common in aerial imagery, while anomalies should still be located. The Gaussian distribution is also a uni-modal distribution. This method is able to model only one modality of the normal class. This is not a problem in the MVTec AD dataset where all training samples are similar and part of the same modality. However, for a general anomaly detection framework where normal images are composed of different normal textures (sea, waves, sun glitters...), this leads to only the majority class being learnt and the minority normal classes being flagged as anomalous. To address these limitations,

we propose a spatially-invariant anomaly localization pipeline using normalizing flows to handle multi-modal normal data.

To build a spatially-invariant anomaly detection pipeline, the anomaly score should not be dependant on the patch coordinates. A simple modification could be to make the model a single Gaussian distribution fit to every patch samples of each image. However, since there are multiple patch modalities, the data may not fit a Gaussian distribution. This can be confirmed by looking at the statistical moments of the patches. Depending on the dimensionality reduction, the skewness and kurtosis of the data are not those of a Gaussian distribution. This is emphasized when using the random dimension downsampling technique proposed in [11]. To use a Gaussian model, we propose to transform the patch distribution into a Gaussian distribution using a normalizing flow (NF). A normalizing flow consists of an invertible transformation $T(\cdot)$ of an unknown input distribution $x = T^{-1}(z)$ to a known latent distribution $z \sim p_Z$. Using the change of variable formula, we can compute the likelihood of any $x$:

$$p_X(x) = p_Z(z) \left| \det \frac{\partial z}{\partial x} \right| \tag{4}$$

where $\det \frac{\partial z}{\partial x}$ is the Jacobian determinant of $T(\cdot)$. Therefore, $T$ is built so that its Jacobian determinant is known and fast to compute. Usually, $p_Z$ is taken to be a centered multi-variate Gaussian distribution $z \sim \mathcal{N}(0,1)$. To perform the transformation $T$, we use the Masked Autoregressive Flow [33] (MAF) model which uses a series of masked autoregressive dense layers, as described in [34]. The masked layers and auto-regressive property allow for a fast probability estimation in a single forward pass. Sampling, however, requires computing a series of probabilities $p(x_i|x_{1:i-1})$ because each $x_i$ is a regression of the previous $i - 1$ variables. In our case, we only leverage the density estimation and do not make use of the sampling from the learnt distribution $p_X$.

The transformation parameters can be trained by maximizing the log-likelihood of the normal only training dataset. Anomaly scores for new samples can be evaluated by computing the negative log-likelihood after transformation of the sample through $T(\cdot)$. Our model is similar to PaDiM estimated using a single shared Gaussian estimator for all patches but with a learnt arbitrary complex transformation of the prior distribution $p_X$ into a Gaussian distribution (see Figure 5). We also experiment with using an ensemblistic approach by using multiple normalizing flows in parallel and by taking the maximum log-likelihood of all models for a given patch. This allows each model to specialize in a type of patch. The loss function for the models is described in Equation (5):

$$\mathcal{L} = \frac{1}{N \times W \times H} \sum_{n,i,j} \min_k \{ -\log p_{Z_k}(T_k(x_{n,i,j})) \} \tag{5}$$

where $k \in \{1, \ldots, K\}$, $K$ is the number of models in parallel, $W$ and $H$ are the dimensions of the patch grid, and $x_{n,i,j}$ corresponds to the embedding vector at location $(i,j)$ of the $n$th sample in the training set. This multi-headed model can also be used to produce pseudo-segmentation maps by using the index of the model giving the highest log-likelihood as a pseudo-label for the patch. Since the normalizing flow used can already model multiple modalities, we found that this modification had little to no positive impact on the performance of the model.

Although anomaly localization models produce anomaly maps, our datasets for animal detection are annotated using bounding boxes. To convert the anomaly map into relevant region proposals, we propose a multi-step pipeline:

1.    First, the anomaly maps are normalized to have their values between 0 and 1;
2.    Then, given an anomaly map $A = \{a_{i,j} \in [0,1]\}$, a threshold $t$ is applied to create a binary map of the same size $A_{\text{bin}} = \{\mathbb{1}_{a_{i,j} \geq t}\}$;
3.    Next, by computing the connected components of this binary map, a set of regions can be proposed using the coordinates and dimensions of each connected component;

4. Finally, using prior knowledge on the dataset, small proposals are removed from the proposed regions. The non-maximum suppression algorithm is also used to filter out duplicate overlapping regions;

5. During the test phase, the proposals are compared to the ground truth bounding boxes using the Intersection over Union (IoU). It measures the relative overlap between two bounding boxes and is commonly used in detection tasks.

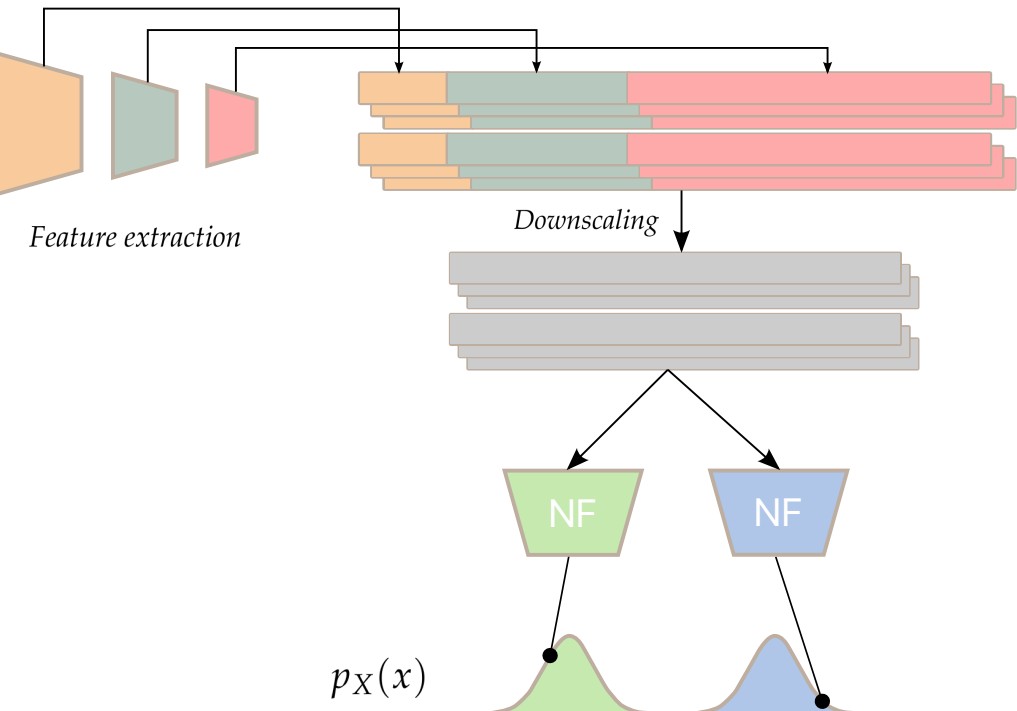

**Figure 5.** Architecture of the multi-headed model. After extracting and downscaling the features, each model computes the negative log-likelihood of each patch and the final score for a patch is the maximum of all $K = 2$ predictions.

The entire box proposal pipeline can be seen on Figure 6.

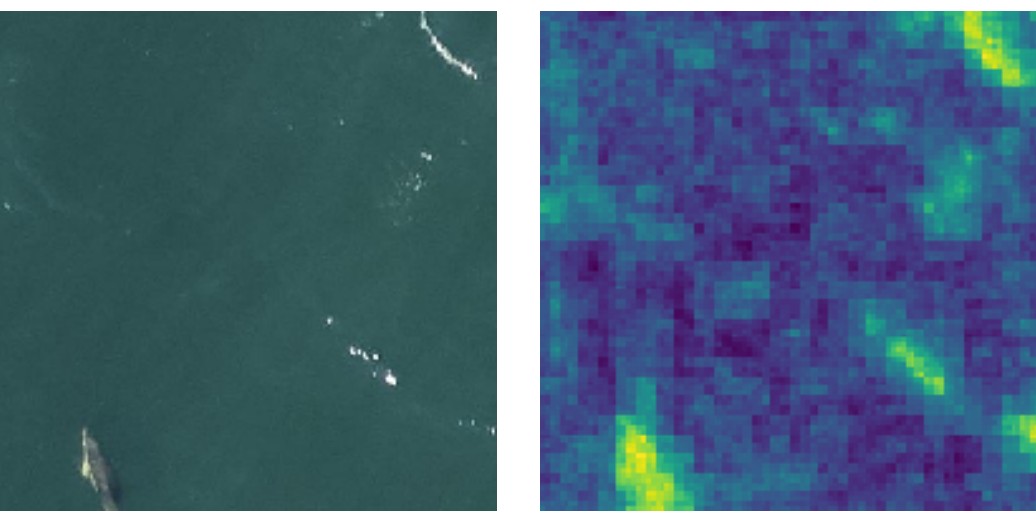

(**a**) Input image            (**b**) Anomaly map (min-max norm)

**Figure 6.** *Cont.*

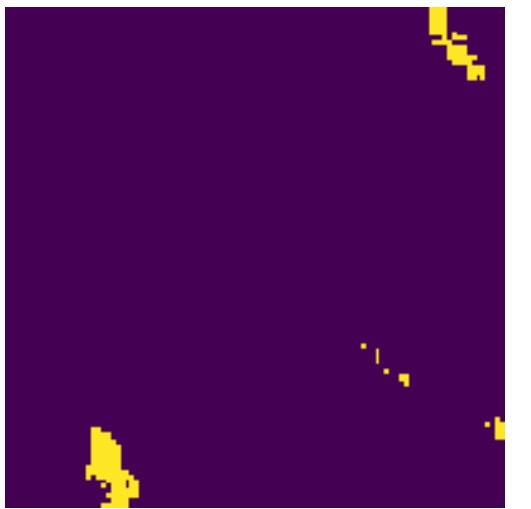 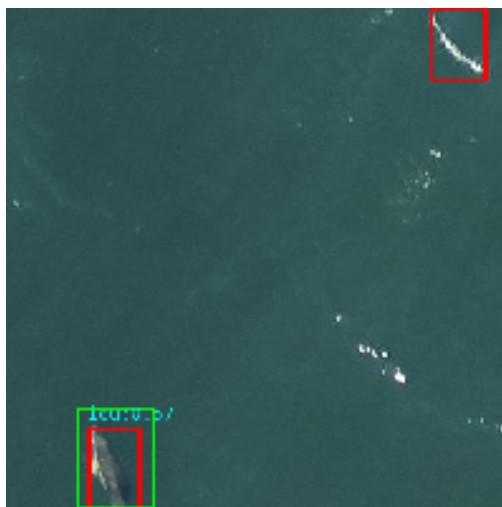

(**c**) Binary threshold ($x \geq 0.8$)      (**d**) Box proposal and IoU

**Figure 6.** Illustration of the box proposal pipeline: First, the input image (**a**) is converted to an anomaly map (**b**) where warmer colors are associated with a higher anomaly score. This anomaly map is then converted to a binary mask (**c**). Finally, the propositions (red boxes) are compared with the ground truth boxes (green) using the IoU (**d**).

## 5. Experiments

### 5.1. Datasets

In this section, we describe the datasets used to evaluate our proposed methods for marine animal detection. Since the aerial imagery is taken with large optical sensors that produce large images, the images are cut into smaller sub-patches of size $416 \times 416$ pixels. When images are cropped into patches, it may happen that one ground-truth bounding box is split into two or four patches. In that case, this ground truth is assigned only to the patch which contains the center of its bounding box. In both datasets, annotations were validated by specialists in marine mega fauna.

**The Semmacape dataset** comprises a set of 165 annotated aerial images acquired as part the SEMMACAPE (https://semmacape.irisa.fr/ (accessed on 4 January 2022)) project, which partially funded the present research and whose main objective is to automatize the survey of marine animals in offshore wind-farms. The images of this dataset were collected in the Gironde estuary and Pertuis sea Marine Nature Park, France, during the spring of 2020. In total, it contains 165 images of 14,204 × 10,652 pixels with 528 ground-truth annotations belonging to one of the following classes:

- *Dolphin* (see some examples in Figures 1 and 2). A total of 258 annotations subdivided into four classes: striped dolphin ( *Stenella coeruleoalba*), common dolphin ( *Delphinus delphis*), common bottlenose dolphin ( *Tursiops truncatus*), and a separated class for dolphins whose species could not be determined;
- *Bird* (see some examples in Figure 4). A total of 270 annotations subdivided into flying and landed birds belonging to four species: gannet, seagull, little shearwater ( *Puffinus assimilis*), and sterna.

Since our focus is on marine animal detection, other classes (seaweed, jellyfish, floating waste, ...) were not included from the testing dataset. The dataset contains a variety of settings from homogeneous sea images to images covered with sun glitters and waves, making it challenging to learn the normal distribution of the data. After filtering and creating the sub-patches, the dataset is composed of 345 patches containing at least one object (anomalous) and 138,544 patches without objects (normal). The percentage of anomalous images is then about 0.25%.

**The Kelonia dataset**, provided by the *Centre d'Etude et de Découverte des Tortues Marines* (CEDTM) and by the Kélonia aquarium (https://museesreunion.fr/kelonia/ (accessed on 4 January 2022)), is composed of aerial images of marine turtles acquired in Réunion island

between 2015 and 2018. This dataset contains 1983 images with ground truth bounding boxes belonging to one of these three classes: *turtle*, *unturtle* (unsure annotations of turtles), and *ray*. In our experiments, we will consider only the turtle and unturtles classes, which comprise the majority of the annotations. Unlike the Semmacape dataset, the images have a larger variety of background and color settings because the sea is shallower, showing the seabed. Furthermore, the training set contains images that may not be representative of the normal class and are not found in the testing set. This makes training on this dataset harder because the learnt distribution may not be optimal for anomaly detection on the testing set. Example samples from the dataset can be seen on Figure 7.

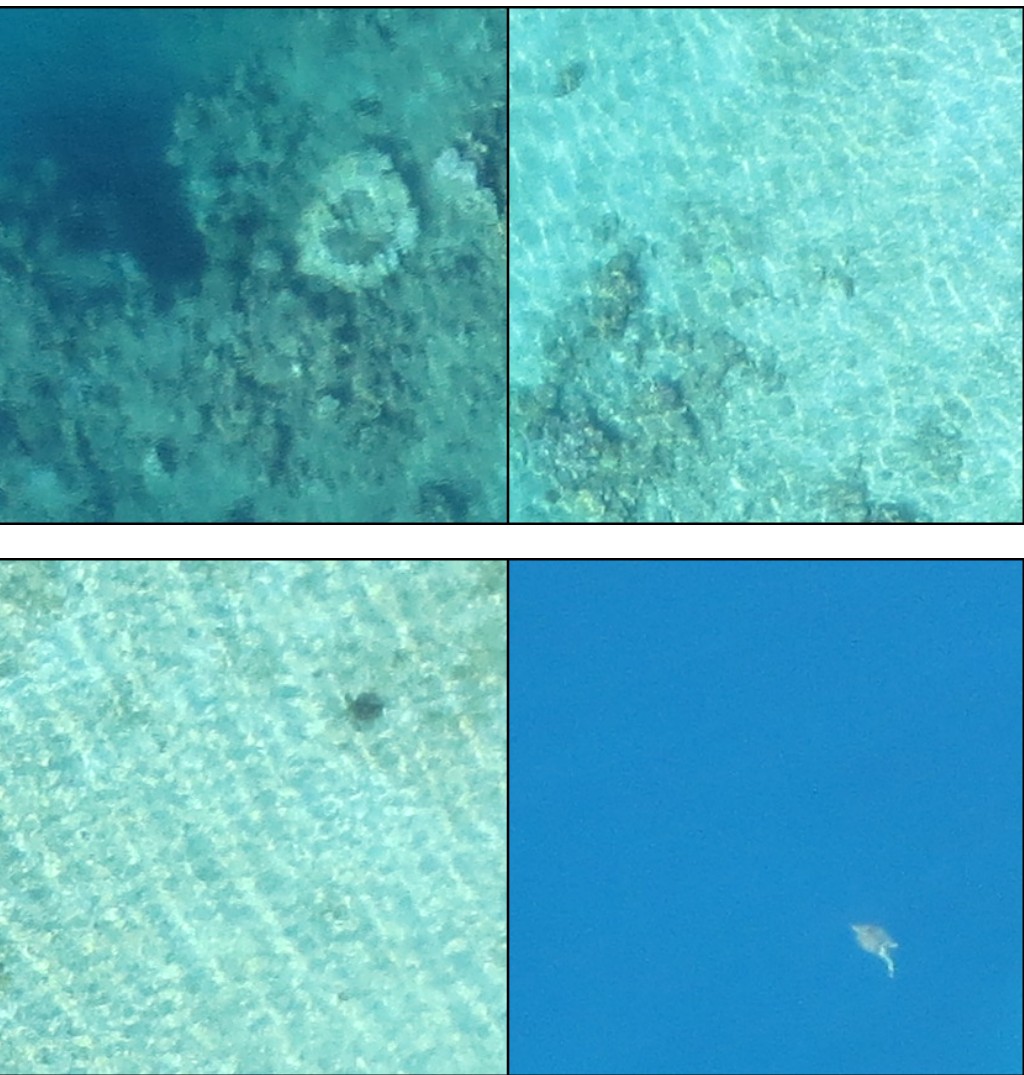

**Figure 7.** Normal (**top**) and anomalous (**bottom**) images from the Kelonia dataset.

The choice of normal images is critical for training an efficient anomaly localization method. Indeed, if the model is trained with only homogeneous images, it will detect sun glitters and waves as anomalies. However, training solely on heterogeneous images will cause the model to expect glitters and waves in a normal setting which leads to unexpected proposals on homogeneous images.

### 5.2. Experimental Setup

As in [11], we use a Wide-ResNet50 [35] as our encoding backbone. The features are then downsampled to a depth of $c = 100$ features using a semi-orthogonal projection matrix as described in [12]. We use seven masked autoregressive density estimator [34]

(MADE) layers in our MAF model. They have seven hidden units with 130 connections each. The Adam [36] optimizer is used with a learning rate of 0.001.

We compare our results with the PaDiM and OrthoAD methods from [11,12] using the same parameters. We also train an adversarial convolutional variational encoder (AnoVAEGAN) similar to [13] to reconstruct normal images. The anomalies are detected by comparing the image reconstruction with the original image using the structural similarity [37] (SSIM) metric. To measure the performance of the detection methods, we consider that a detection is positive if the IoU between the prediction box and the ground truth box is greater than 0.1. The F1 score, recall and precision can then be measured. They are computed as follows:

$$\text{Recall} = \frac{\#\text{detected}}{\#\text{objects}} \tag{6}$$

$$\text{Precision} = \frac{\#\text{detected}}{\#\text{proposals}} \tag{7}$$

$$\text{F1 score} = \frac{2 \times \text{Recall} \times \text{Precision}}{\text{Recall} + \text{Precision}} \tag{8}$$

Because the F1 score blends information about both the recall and precision, we use it as our main metric. We also evaluate the classification performance between anomalous and normal images of the models by computing the area under the receiver operating characteristic curve (AUROC). The anomaly score for an image is defined as the maximum anomaly score among all its patches.

*5.3. Results*

The object detection scores for the Semmacape and Kelonia datasets are given in Tables 1 and 2, respectively. For all metrics, higher scores indicate better performance. On both datasets, the highest F1 scores among all tested approaches were obtained by one of our proposed methods. The most significant improvements were observed on the Semmacape dataset, for which our method provided an improvement of 6.1% and of 22.6% in terms of F1 and recall scores, respectively, with respect to the state-of-the-art AnoVAEGAN [13]. On this dataset, the classification of patches into anomalous and normal images is also significantly improved by our method, as attested by an augmentation of 12.4% of AUROC in comparison to OrthoAD [12]. On the other hand, more modest improvements were observed on the Kelonia dataset: 1.3% and 5.2% in terms of F1 and recall scores, respectively, when compared to OrthoAD [12].

The improvement from using a normalizing flow to transform the embedding vector is greater on the Semmacape dataset than on the Kelonia dataset. This is due to the fact that the training dataset for Semmacape contains normal patches that are different from anomalous patches of the testing set. For the Kelonia dataset however, there are rocks and patches that are more similar to the turtles in the training set. Since the negative log-likelihood is minimized during training, some "turtle" patches will then be assigned a high likelihood because they are similar to rocks in the training set. This can lead to some failed detection because the area around the object will have a normal score. For the same reason, the generative method AnoVAEGAN performs worse on the Kelonia dataset because of training images containing rocks that are similar to turtles. The model is then able to reconstruct the turtles accurately and fails to detect them during testing. The classification performance on the Kelonia dataset can also be explained by the fact that some normal images are very dissimilar from the rest, which leads the model to classify them as anomalous.

**Table 1.** Results on the Semmacape dataset. All the models have been trained using the same set of 1000 training images.

| Method | F1 Score | Recall | Precision | AUROC |
|--------|----------|--------|-----------|-------|
| PaDiM [11] | 0.383 | 0.434 | 0.343 | 0.606 |
| OrthoAD [12] | 0.458 | 0.373 | 0.594 | 0.795 |
| AnoVAEGAN [13] | 0.469 | 0.531 | 0.420 | 0.697 |
| Ours, 1× MAF [33] | 0.530 | 0.757 | 0.408 | 0.919 |
| Ours, 2× MAF [33] | 0.486 | 0.523 | 0.455 | 0.869 |

**Table 2.** Results on the Kelonia dataset. All the models have been trained using the same set of 1000 training images.

| Method | F1 Score | Recall | Precision | AUROC |
|--------|----------|--------|-----------|-------|
| PaDiM [11] | 0.504 | 0.443 | 0.586 | 0.431 |
| OrthoAD [12] | 0.571 | 0.514 | 0.643 | 0.431 |
| AnoVAEGAN [13] | 0.051 | 0.033 | 0.107 | 0.469 |
| Ours, 1× MAF [33] | 0.568 | 0.559 | 0.578 | 0.410 |
| Ours, 2× MAF [33] | 0.584 | 0.566 | 0.604 | 0.391 |

When varying the IoU threshold for positive predictions (Figure 8), the F1 score decreases as the threshold increases. With an IoU threshold of 0.3, the F1 score is down to less than 0.45 for the best performing method on the Semmacape dataset. A higher IoU threshold means that the proposed bounding boxes must be more similar to the ground truth boxes in order to be counted as a positive prediction. For the PaDiM method, the rate at which the performance decreases as the IoU threshold increases is greater than our method. This is because large objects can sometimes be counted as positive even with failed predictions when the anomaly threshold is low and a large portion of the image is proposed as a region of interest. This region will then have an IoU greater than the IoU threshold with the ground-truth bounding box. We can see that the PaDiM method has a larger confidence interval on the F1 score because of the random dimension selection which plays a role in the performance of the model and is effectively sampled differently at each run.

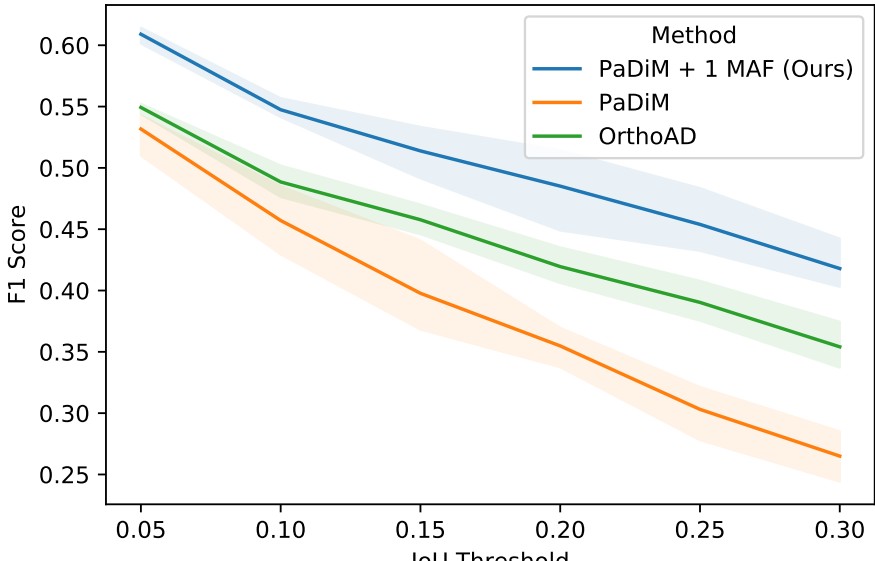

**Figure 8.** F1 score on the Semmacape dataset with different IoU thresholds. The mean performance and 95% confidence interval is reported over 3 runs.

As seen on Figure 9, the methods are not very sensitive to a variation in the number of training samples. This means that modeling the normal dataset does not require a large number of training samples. In fact, since some images have similar visual features because they are shot in sequence above the sea, having a wide variety of images covering the whole spectrum of the normal setting is more important than having many similar training samples. However, too few training samples can lead to over-fitting of the model which will cause new normal images to be classified as anomalous during testing because they are different from the training set. In practice, we train with 1000 training images. Examples of predictions for both datasets can be seen on Figure 10.

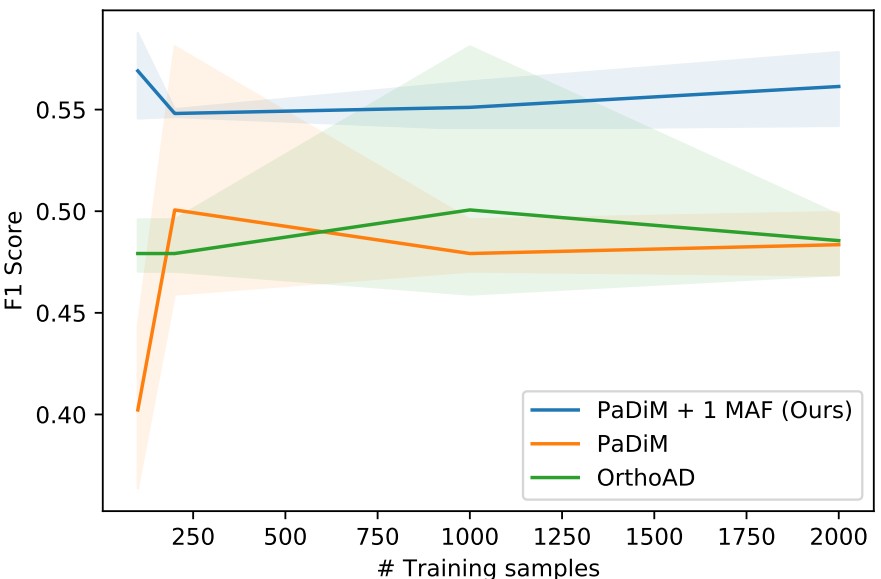

**Figure 9.** F1 score on the Semmacape dataset with different numbers of training samples. The mean performance and 95% confidence interval is reported over 3 runs.

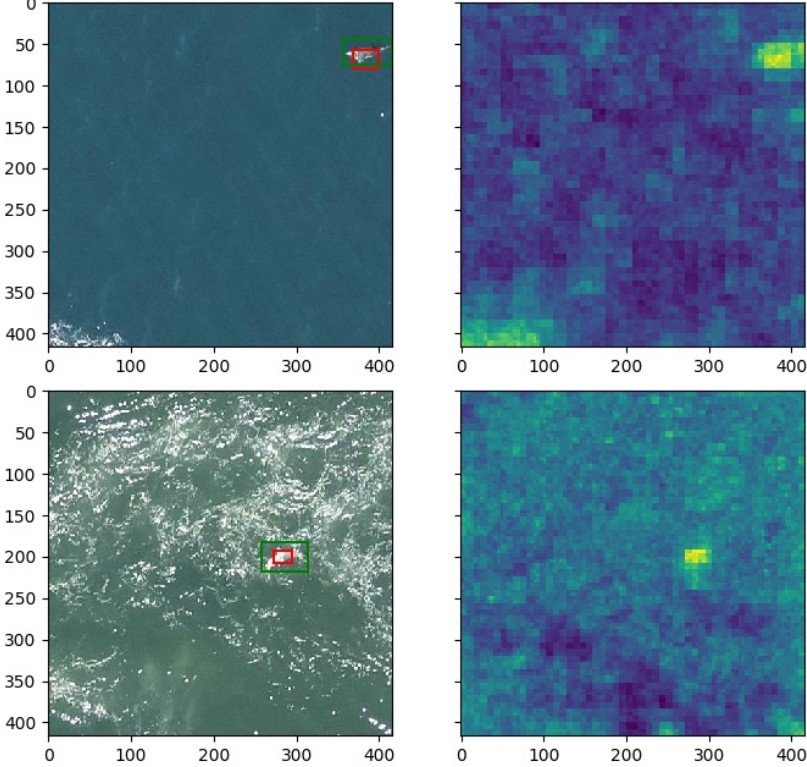

**Figure 10.** *Cont.*

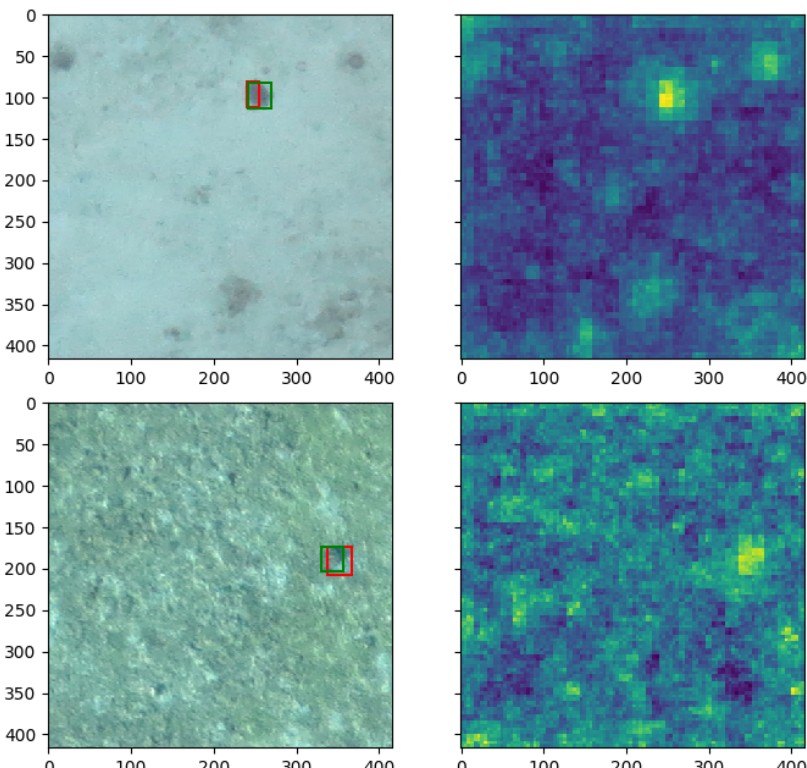

**Figure 10.** Example predictions (**left**) and their corresponding anomaly maps (**right**) from the Semmacape (**2 first rows**) and Kelonia (**2 last rows**) dataset. Rocks and waves have a higher anomaly score than water, using the appropriate anomaly threshold is important for the proposed regions to be interesting.

## 6. Discussion

Our method proposes a first step in performing weakly supervised marine mammal detection. As such, it requires less human intervention on the training data generation process compared to supervised methods and is also able to propose individual bounding boxes for each detected mammal. However, in this study we do not consider the problem of classifying species. As a consequence, other anomalous floating objects such as boats or marine debris could be detected. In practice, the performance from our method should be assessed and the predictions should be confirmed by comparing the output to other population estimation methods, such as field sampling campaigns.

The proposed bounding boxes could also serve as a starting point in computer assisted ecology by guiding human annotators to larger area of interest first or as an initialization method for other computer assisted annotation methods which require pre-trained models or active learning [38].

## 7. Conclusions

By transposing the problem of anomaly localization from an industrial setting to marine animals localization, we are able to provide class-agnostic bounding box proposals on aerial imagery. The produced detection can either be used to speed up the object discovery for new flight surveys or for direct bounding box proposal and animal population density estimation. By leveraging pre-trained convolutional neural network features without full annotations, the proposed approach is able to detect marine animals. Although not yet on par with supervised methods, this is a first step on enabling weakly supervised detection of marine animals. During the work, one of our observations is that the training set should contain normal samples with good quality, since most anomaly detection methods, including ours, are often sensible to the contamination of the training set with anomalous images.

Despite its great potential in marine animal localization from aerial images, the proposed method cannot classify between different species. Future work on unsupervised clustering of proposals could result in improving precision by detecting irrelevant proposal beyond providing some solutions for the unsupervised classification task.

**Author Contributions:** Conceptualization, P.B., D.S.M., M.-T.P. and S.L.; methodology, P.B., D.S.M., M.-T.P. and S.L.; software, P.B.; validation, P.B.; investigation, P.B.; writing—original draft preparation, P.B and D.S.M.; writing—review and editing, P.B., D.S.M., M.-T.P. and S.L.; supervision, D.S.M., M.-T.P. and S.L.; project administration, M.-T.P. and S.L.; funding acquisition, S.L. All authors have read and agreed to the published version of the manuscript.

**Funding:** This work was supported by the SEMMACAPE project, which benefits from an ADEME (*Agence de la transition écologique*) grant under the "Sustainable Energies" call for research projects (2018–2019).

**Institutional Review Board Statement:** Not applicable.

**Informed Consent Statement:** Not applicable.

**Data Availability Statement:** The Semmacape and Kélonia datasets used in the present research are not publicly available.

**Acknowledgments:** The authors would like to thank the anonymous reviews for their suggestions, and the *Centre d'Etude et de Découverte des Tortues Marines* (CEDTM) and Kélonia aquarium whose data was made available for the present research.

**Conflicts of Interest:** The authors declare no conflict of interest.

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
