# Peer review of "Weakly Supervised Detection of Marine Animals in High Resolution Aerial Images"

_remotesensing, doi:10.3390/rs14020339_

Round 1

Reviewer 1 Report

After reading the paper I really like it and the methods presented seem very innovative and can find an audience in application. I don't want to force some comments because in my opinion the paper can be published as i.

Author Response

Dear Reviewer,

Please find our response to your comments in the attached pdf file.

Kind regards,

Reviewer 2 Report

Dear Editor,

Dear Authors,

I finished reading the new version of the manuscript number remotesensing-1488144, entitled "Weakly supervised detection of marine animals in highresolution aerial images" written by  Berg et alii, concerning a better adapted remote sensing approach for marine animal detection.

I have carefully read the revised manuscript with interest and I appreciated very much the its contents. I actually found the manuscript worthy of publication but not in its present form; some parts of the text requires special attention in reading, making it more fluent. I am not a native speaker but the language seems so scholar in different sections. I believe some English colleague may provide a contribution, but I leave the Editor decide if the language requires a further revision before the manuscript final publication.

My main criticism regards Conclusion. Sepcifically, I would strongly emphasize the usefulness of a confirmation of the data analyzed in this study through a field sampling campaign aimed at verifying the actual presence of cetaceans in the analyzed area using marking methods. It is true that these studies are noteworthy, but without any type of confirmation there is a risk of underestimation of cetaceans so that managers may to be driven to incorrect management actions.

Good lack and stay strong.

Author Response

(The authors gave the same response as above.)

Reviewer 3 Report

This article on supervised detection of marine animals from aerial images in an important contribution tot he field and is suitable for the journal. Overall this is a straightforward well written paper and was a pleasure to read. Also the authors do not overstate their results (that they could not clarify between species) but make an excellent start for others to apply and improve this method. I make some small suggestions for improvement.

Abstract: the results are not really presented in the abstract - only weakly in the last line; discussion is also missing.

Introduction: I would expect to see more references in the introduction - indeed in the first three paragraphs there is only a single reference about offshore wind energy. This needs to be expanded. For example, in paragraph two, there are numerous references on learning models that could be cited here.

Good practice to publish the codes online and this paper is likely to be cited often!

Figures 1-4 are very useful and interesting and help to illustrate the methods

Section 3 is clear and excellent

Lines 208-280 are very clear and the figure is very helpful - this study/method is highly replicable

Page 9 - line 207 - the genera in this section should all be upper case

Figure 10 should not in the references

Author Response

(The authors gave the same response as above.)
